# Virtual Power Plants Optimization Issue: A Comprehensive Review on Methods, Solutions, and Prospects

Wafa Nafkha-Tayari [1], Seifeddine Ben Elghali [1,*], Ehsan Heydarian-Forushani [2] and Mohamed Benbouzid [3,4]

1  Laboratory of Information and Systems (UMR CNRS 7020 LIS), Aix Marseille University, 13397 Marseille, France; wafa.nafkha@lis-lab.fr
2  Department of Electrical and Computer Engineering, Qom University of Technology, Qom 1519-37195, Iran; heydarian@qut.ac.ir
3  Institut de Recherche Dupuy de Lôme (UMR CNRS 6027 IRDL), University of Brest, 29238 Brest, France; mohamed.benbouzid@univ-brest.fr
4  Logistics Engineering College, Shanghai 201306, China
*  Correspondence: seif-eddine.ben-elghali@univ-amu.fr

**Abstract:** Recently, the integration of distributed generation and energy systems has been associated with new approaches to plant operations. As a result, it is becoming increasingly important to improve management skills related to distributed generation and demand aggregation through different types of virtual power plants (VPPs). It is also important to leverage their ability to participate in electricity markets to maximize operating profits. The present study focuses on VPP concepts, its different potential services, various control methodologies, distinct optimization approaches, and some practical implemented real cases. To this end, a comprehensive review of the most recent scientific literature is conducted. The paper concludes with remained challenges and future trends in the topic.

**Keywords:** virtual power plant; optimization approaches; VPP services; microgrid

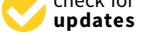



## 1. Introduction

Around the world, renewable energy investors have taken advantage of the strong development of distributed energy resources (DER). Growth in DER integration globally is around 20% by the end of the 20th century [1]. A microgrid is a small-scale ("micro") power grid that enables either isolated or grid-connected operation modes independently or in conjunction with the region's main power grid. Different types of microgrids exist,and obviously, many definitions also exist. The most common definition, given by [2], reads as follows: "A microgrid is a group of interconnected loads and distributed energy resources within clearly defined electrical boundaries that acts as a single controllable entity concerning the grid". The microgrid concept has become widespread over the last decade, and many microgrids are functioning in different parts of the world. The grid-connected mode offers the microgrid the benefits of power trading with the main grid. However, in case of disturbances or failure in the main grid, its operation shifts to an islanded mode. In both cases, operators must deal with DERs variability and uncertainties. As a result of these troubles, research areas have been have been established to find appropriate solutions for the successful integration of DERs while ensuring power grid reliability and stability.

Within this context, the virtual power plants (VPP) was conceived. Although there is a widely held simple interpretation of VPP in the narrow sense: "A virtual power plant assembles a multitude of low- and medium-power generation units, energy storage devices, and remotely controlled flexible loads to participate in the electricity market or provide complementary energy services, including balancing, to grid operators". According to their points of view, authors in the literature propose two other definitions: one with a commercial aspect [2,3] and another with a technical aspect [2,4–6]. By way of example,

in [6], VPP is defined by its role in controlling DER, flexible loads, and storage as an information and communication system. The study in [6] states that the VPP including DER, controllable loads, and aggregate storage units is operating as a single power plant managed by EMS. In [2], in order to realize contracts in the wholesale market, the VPP is defined as a trading platform of DERs.

The VPP concept is highly recommended to mitigate the negative drawbacks of DERs integration. VPP ensures the aggregation of all DERs as a single power plant providing more flexibility to the grid to enhance its reliability and stability, without overlooking the many other benefits and opportunities for consumers, producers, and network operators. VPPs contribute to the reduction of carbon emissions through the aggregation of renewable resources that lessen dependence on fossil fuels. On the other hand, VPPs, equipped with complementary data on climate and local geography, take part in mitigating the impact of urban density on the performance of distributed renewable systems. In addition, VPPs can also reduce the negative environmental impact and improve the security of renewable energy through the optimal management of these distributed units [2].

In [7], the main differences can be listed as follows:

- The VPP continues to operate normally in the event of a single user failure, while this problem affects all connected sub-systems of the microgrid.
- The advantage of VPP over microgrids is that the former uses less ESS than the latter. Thus, VPP is less costly to implement and offers a coherent solution.
- For MG implementation, all the resources must be in a geographic area; however, VPP and particularly commercial ones facilitate having a single entity on behalf of a set of resources not related to the same geographic location.
- VPPs use artificial intelligence to develop simple algorithms that ensure optimal production and consumption, unlike microgrids that use very complex optimization algorithms.

Despite these differences, the migration from microgrid to VPP is possible and can be easily achieved through the smart grid concept, which has already been tested in some countries.

The management of production, data transmission, distribution, and demand control is based on intelligent algorithms that use communication protocols such as the Internet Protocol. With the incorporation of IoT, the communication and behavioral technique of each DER is similar to a neural network. The Web-to-Energy project is one of the biggest developments in the field of smart grids and can be readily adapted to the VPP concept. Figure 1 shows the evolution of a microgrid into a VPP and how cells are clustered and connected to a centralized VPP. These aggregators interact with each other and operate as a single entity. The localized control center behaves as a self-organizing intelligent solution.

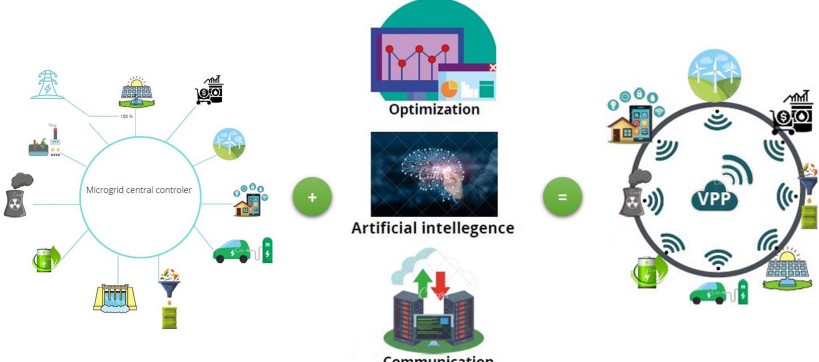

**Figure 1.** Evolution of MG to VPP.

To make the paper easier to understand and read, Figure 2 illustrates the step-by-step information flow of each section. This article presents the concept of VPP, its different services, a variety of control methods, and a comparison of different optimization methods.

These are classified according to their control method, system modeling, and type of problems addressed.

Therefore, the significant contributions of the paper can be recounted as follows:

- This article discusses the concept of VPP in its entirety. It consists of presenting the different notions from its architecture to the different types of electricity market.
- Classify the different internal control methods and the main optimization algorithms used for each.
- To explain optimization strategies for VPP based on system configuration, parameters, and control techniques.
- To summarize differents markets.
- To give examples of the latest practical implementations of VPP.

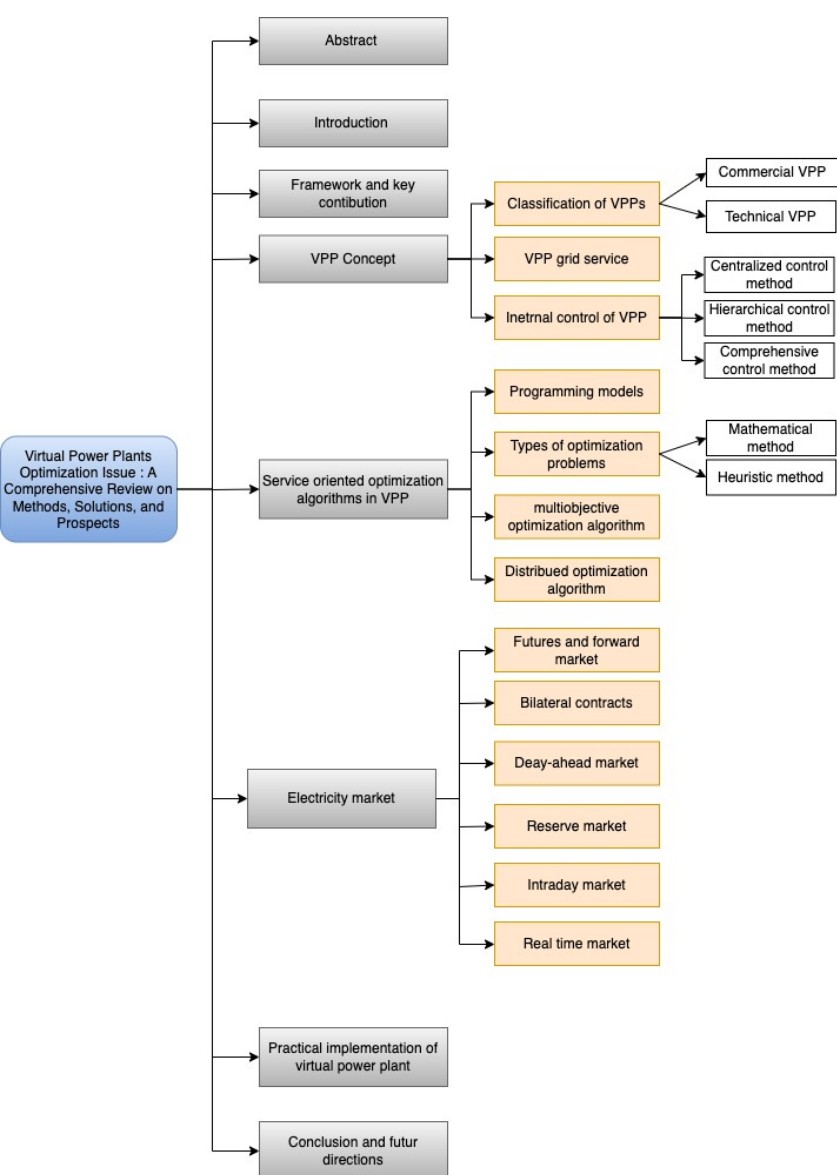

**Figure 2.** Paper structure.

## 2. VPP Concept

The concept of VPP was derived from the virtual utility framework from Awerbuch and Preston in 1997 [4,8]. The virtual utility is based on a flexible cooperation between utilities to provide high-performance electric services to customers through the virtual sharing of their private assets. The above concept improves the efficiency of individual utilities

and avoids redundant constructions. In order to evolve the original definition of virtual utility, the concept of VPP is mainly based on the mathematical combination of different energy resource cost curves to form a virtual aggregation of DERs. As shown in Figure 1, VPP changes the traditional power system topology through establishing an information exchange among variable energy resources and coordinating their generation profiles. Virtual Power Plants (VPPs) are used to optimize the management of a generation fleet using a control center that can remotely manage the generation, load shedding, and storage resources within its perimeter. In addition, VPPs can be used to dynamically optimize energy costs by reacting to market price variations (Commercial VPP) or contributing to the electrical system's supply/demand balance by responding to signals from network operators (Technical VPP) such as a microgrid.

### 2.1. Classification of VPPs

Constrained by distance, capacity, size, and resource types, the VPP operates within a collaborative system called the Internet of Energy (IOE) composed of different layers (i.e., Information and Communication Technology (ICT) and Advanced Measurement Infrastructure (AMI), ...) as shown in Figure 3. The main objective of the VPP is to obtain the optimal benefit by recognizing the information associated with the demand and supply of energy. VPPs can be grouped into two broad categories: Commercial VPP (CVPP) and Technical VPP (TVPP) [2].

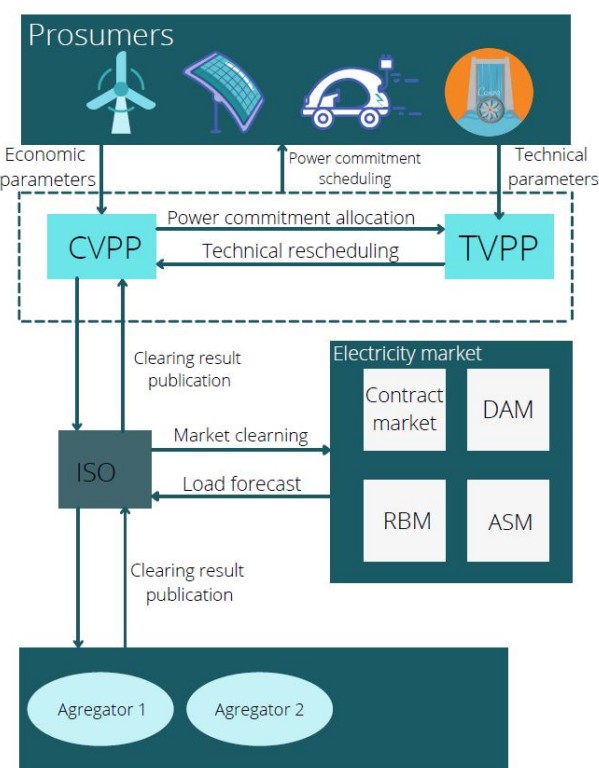

**Figure 3.** Framework of VPP.

### 2.1.1. Commercial VPP

The main objective of the CVPP is economic optimization. This involves financial risks: costs, optimized revenues for energy exchange, the combination of economic paradigms with smart grid services, and representation of supply and demand tables. Risk management methodology is generally the basis of studies in this area. The developed models deal with agreements between VPP users, partnership contracts with distribution companies, and mutual agreements between different VPP groups, as well as energy marketing issues. In a CVPP, the aggregator is the price negotiator for its electricity production with the electricity market operators. A CVPP is primarily based on economics so that the aggregator can

reach the most beneficial agreement among the many aggregators in the market. This commercial aggregation is not dependent on grid operation. The CVPP provides visibility and input from multiple DER units into the energy markets. Its objective is to quote supply and demand in the wholesale energy market while balancing trading portfolios. Unlike standalone DER units, it reduces capacity imbalance by integrating many smaller units. The CVPPs also provide demand-side management in the event of an outage [9,10]. Additional tasks of CVPPs also include demand and production forecasting, DER characteristics analysis, market bidding, and generation schedules to maximize profits.

### 2.1.2. Technical VPP

DERs in a TVPP are grouped in the same geographical area. The main role of the TVPP is to incorporate the influence of the local network in real time, as well as the representation of the costs and operating characteristics of the portfolio. In a TVPP, all relevant and accurate data regarding the power generation of each DER unit and future generation statistics come from the aggregator. Furthermore, the aggregator needs all the information regarding the generation power profile and forecasting algorithms to manage the TVPP better. Local system management for a Distribution System Operator (DSO), as well as ancillary services for a Transmission System Operator (TSO) and system balancing, are part of the services and functions of a TVPP. A TVPP continuously monitors the status of individual loads and manages assets based on statistical data. Furthermore, it can locate faults and help repair them. It also offers statistical analysis and project portfolio optimization functions [9,10]. In addition to fast metering service, TVPP also offers local monitoring service and monitoring of batteries and inverters. It handles complex calculations, technical applications, storage, and optimization [9]. The research studies are mainly concerned with financial issues, monitoring, and fault detection. This classification involves energy flow optimization, technical feasibility solutions, communication protocols in smart grids, and some fuzzy algorithms related to production and consumption. In addition, security is one of the main issues, as VPPs need to ensure the security of their personal information. One of the most worrying issues of a smart system is the animosity of cyber-attacks and viruses, as consumers' personal information must be protected in case of cyber-attacks. Therefore, the TVPP takes care of this situation. It becomes an essential part of the communication system [10].

### *2.2. VPP Grid Services*

The VPP can improve grid performances by providing several services:

- Localized clean energy: a VPP unit helps to achieve global warming and pollution. It integrates new technologies and methods that ensure the production of clean energy.
- Real-time demand response: the neural network established between the renewable energy sources (RES) ensures a real-time response to the demand. VPP technology brings all resources' features and keeps the system in balance.
- Frequency and voltage control: The VPP allows to better manage sudden frequency changes with a fast and efficient response. Moreover, the peak demand is better controlled thanks to the control of the energy flow. It draws the necessary energy from the sources of the nearest neighbors.
- Big data from small sources: Transforming utilities into a digital network can result in high performance by managing the considerable accuracy of the data.

### *2.3. Internal Control of VPP*

According to the literature, internal control methods can be divided into centralized control, distributed control, and comprehensive control. In the following, the characteristics of each control method are discussed.

### 2.3.1. Centralized Control Method

The centralized control method allows VPP to have full control authority. In [11,12], the VPP establishes a control coordination center that guarantees the absolute authority to distribute all integrated DERs. The communication network that brings together internal and external data for VPP decision making is the main factor determining the control topology of the centralized control method. In addition, with an efficient communication channel, the power scheduling of the VPP can be transmitted to the production terminal in real time. Thus, the DERs are better integrated through the centralized control method, and the initially uncoordinated low-capacity DERs become large-scale production collaborators. Furthermore, as discussed in [13,14], under its considerable influence on the market-clearing procedure, the VPP is considered a price provider in the electricity market by centrally coordinating large-scale generation capacity. Nevertheless, in [15,16], the VPP has a small capacity and limited influence, defining it as a price taker. The expanding influence of VPPs on the functioning of the power system and the electricity market is increasingly recognized. However, the role of VPPs in the market may vary depending on their embedded resources. The computing power of the centralized control method continues to multiply rapidly. Indeed, multiple variables have to be considered during the optimization procedure, since the VPPs determine all the operating profiles of DERs in this type of control methodology. This leads to an increase in the difficulty of the computational task of the VPP control center. As mentioned in [17], the mixed integer nonlinear programming (MINLP) model is typically used as an internal centralized control model. It can also be transformed into a mixed integer linear programming (MILP) model by a dualistic transformation or Karush–Kuhn–Tucker (KKT) optimality condition. Then, the two-layer optimization problem between VPP and ISO can be converted to a single-level problem. The authors of [18] used stochastic programming with a predefined uncertainty probability distribution for risk analysis. In [15–19], the MILP formulation is used for the VPP centralized control problem, where a robust optimization technique is applied. The latter considers the worst-case scenario compared to stochastic programming, which improves the reliability of the optimization results.

In [11,12], a heuristic algorithm such as particle swarm optimization (PSO) or genetic algorithm (GA) is applied to solve the centralized control problem of VPP. The modeling of the PPV, in this case, is to consider it as a unique problem for which the application of the mathematical model is forbidden because of the inter-temporal constraints. As shown in [20,21], the CPU problem is transformed into a security constraint unit commitment (SCUC) engagement problem, where modifications are applied to the original heuristic algorithms to accelerate computational efficiency by introducing network topology constraints. However, this methodology could facilitate the coordination of DERs to provide more services because the resources are managed centrally. However, it needs more computation capacity, storage space, stronger solvers and processors, and a communication platform. The other drawback of this method is related to its vulnerability to the possible communication problems or even cyber-attacks.

In conclusion, in the centralized control method, the VPP aims to optimize the scheduling of internal energy resources, where sufficient bandwidth is the foundation for information retrieval and good decision making. In [22], bandwidth was estimated at 3.94 kBytes/s for normal functioning and data flow (e.g., requests and count reports) in both directions for a data attribute and a single DER. With the addition of data attributes (timestamps, multiple metering tags, etc.), bandwidth peaks can be as high as 80.5–97.5 kBytes/s) and with each DER (VEN) added, bandwidth increases. However, the centralized control of VPPs is limited to a reasonable scale, which is constrained by design efficiency.

### 2.3.2. Hierarchical Control Method

The structure of the distributed control method consists of two independent levels. The first level is managed by the VPP, creating the central communication level. In contrast,

the second level is managed by the Distributed Energy Resources (DERs), forming the independent subsystem level as shown in Figure 4.

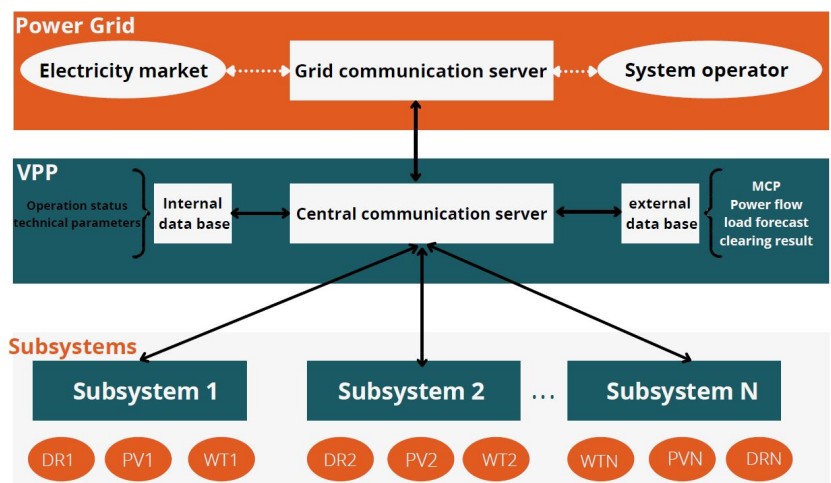

**Figure 4.** VPP distributed method.

As presented in [23,24], the independent subsystem plans the local energy resources to maximize individual profit. The VPP has the exclusive authority to manage resources through an information and communication service between multi-subsystems. The monopoly problem in the market of large-scale centralized VPPs is thus avoided by applying a distributed control method. In addition, the massive computational burden is alleviated by distributing the decision variables among the subsystems. On the other hand, recent research has highlighted the potential drawbacks of distributed control. Indeed, the lack of centralized optimization can lead to a contradiction between the operating profiles of each subsystem, which decreases the overall competitiveness and worsens the competition in the market. As discussed in [23], the combination of autonomous operating strategies is computationally highly demanding. However, game theory can be applied to accelerate the convergence of uncoordinated strategies. In addition, distributed computing algorithms are also employed to address the distributed control problem. In [25], the subsystem is conceived as an agent, whose main objective is to achieve individual benefit maximization. In [26], the authors proposed a new control strategy known as the symmetric component control algorithm to operate distributed energy systems. In [27], some constraints such as delays, channel noise, and channel faults have been added to the non-ideal communication network. Applying the distributed primary and dual subgradient algorithm improves both the communication and computational efficiency in the distributed scenario.

### 2.3.3. Comprehensive Control Method

In [28,29], the authors proposed a comprehensive control method that combines the benefits of centralized and distributed control. This control method can be divided into two secondary control levels so that the secondary levels are closely related to the VPP control center.

- Centralized VPP Control: The agents' bidding strategies are coordinated centrally by the VPP to form a final market participation strategy. This alleviates the computationally intensive nature of the centralized control method. In fact, it is distributed among the agents in level II.
- Distributed Agent Control: A local optimization is handled by the distributed agents who submit their operation profile to the VPP for global optimization. As the VPP issues the final coordinated operation profile, all agents proceed with the regional reorganization and execution.

### 3. Service-Oriented Optimization Algorithms in VPP

Ensuring the reliability and security of the electricity generation and transmission system is the primary objective of the auxiliary services market. Ancillary services ensure the balance of generation and demand in the system. On the one hand, from an economic point of view, the participation of VPPs is expected to increase significantly due to the increase of liquidity and competition in the ancillary service markets. On the other hand, from a technical point of view, the progressive proliferation of renewable generation facilities in current power systems may fragilize them, as a grid collapse may result from mismanagement, which cannot ensure the reliability of power generation. Therefore, several studies have included participation in ancillary service markets in the modeling of VPPs to allow for frequency-power control that ensures quality and security of electricity supply [18,30–40]. These VPP models are equipped with storage systems. These are essential to mitigate grid stability issues that may arise due to variations in renewable energy generation. In the VPP optimization framework, one critical step to maintaining and balancing demand and production is accurately predicting energy production and consumption. As prices vary with the evolution of demand over the installed capacity, the goal is to lower costs and achieve a more sustainable and balanced energy flow. To achieve this, the AI-based algorithm is becoming popular. On the other hand, weather conditions are crucial for renewable energy resources such as photovoltaics and wind power. Therefore, elaborated mathematical methods, physics and specialized software algorithms are required to accurately predict them. The developer updates the optimization and forecasting algorithms whenever the state of the power system changes (which may include a new power supply). In some cases, the developer must go through a completely different algorithm.

*3.1. Programming Models*

The most commonly used models in the technical literature are listed in Table 1.

**Table 1.** Programming models on VPP.

| Proposed Model | Optimization's Goal | Contributions | References |
|---|---|---|---|
| Stochastic programming and robust optimization | Maximize the VPP profit | Combination of bilateral contracts and the DA market | [31,41–43] |
| Static robust optimisation | Deal with uncertain wind power production and market prices | Proposed stochastic adaptive robust mixed integer linear programming | [17,30,44,45] |
| Robust stochastic models | Improve the prediction accuracy | Energy management method that controls the model prediction by modifying the time scale so as to significantly improve the prediction accuracy | [46–48] |
| Stochastic hybrid intelligent algorithm | Prediction of energy storage unit's·level | Balanced energy and cheaper priced electricity have been obtained | [49–51] |

Uncertainties through prediction intervals are represented by robust models. A robust model is presented in [41,52]. It provides energy management decisions that minimize the most adverse cost due to uncertainty sources. In addition, if the uncertain data take values from the prediction intervals considered, then the robust model ensures that all constraints are met. In contrast, stochastic programming methods use a limited number of scenarios to model uncertainties. In addition, the expected cost on the achievements of the considered scenarios is minimized by the solutions provided by a stochastic programming model [44]. Thus, it is generally necessary to take into account a sufficiently large number of scenarios. A robust stochastic model combines the two previous models. It uses both prediction intervals and a set of scenarios to represent uncertainties. The combined optimization approach can compute significantly quicker than stochastic optimization, deal with risk

better compared to deterministic optimization, and have much better performance in probability than robust and interval optimization. In addition, the combined optimization can handle the uncertainties of the PPV in a limited computation time.

- The robust model offers a better performance from a computational point of view. This asset allows it to be used optimally in the RT decision processes.
- In most cases, the profitability of the robust model is higher than that of the other two models.
- The robust stochastic model is more computationally efficient than the stochastic model, although the latter has a better economic performance.

### 3.2. Types of Optimization Problems

Optimization problems can be classified into the following categories according to the type of variable (continuous, integer) and the linearity or not of the constraints:

- Linear Programming (LP);
- Mixed-integer linear programming (MILP);
- Nonlinear programming (NLP);
- Mixed-integer nonlinear programming (MINLP).

Some of optimization problemsare listed and detailed in Table 2.

**Table 2.** Optimization problems on VPP.

| Optimization Problem | Optimization's Goal | Contributions | References |
|---|---|---|---|
| An adaptive robust approach MILP | The optimal DA (or RT) energy, reserve dispatch and the worst-case realization of uncertain DA (or RT) market energy prices | Achievable for all possible cases of the considered uncertainties within a confidence limit and also optimal for the worst case realization of these uncertainties | [18,29,44] |
| Mixed linear integer method | Forecasting wind speed and solar radiation | By using a prediction algorithms, occured faults can be detected by comparing obtained results with power generation | [12,13,16] |
| CPLEX (IBM Log Optimisation Studio) program based on mixed-integer linear programming | Results of real-time integration of DER into VPP | Optimal real-time integration of VPP improves economic feasibility and produces reliable and functional power. Research studies are being conducted on how grid extension and the grid can affect feasibility and performance. | [31] |

The mixed linear mathematical model is formulated using integer decision variables associated with the hourly import/export of electricity during the scheduling period or the state of charge/discharge of the storage systems, among other factors, in addition to continuous variables that correspond to the values of the energy exchanged in the VPP model. Simplicity and speed in finding the optimal solution are the main advantages of this model. The optimal solution of this model can be easily obtained with a software with an efficient solver. On the other hand, in the literature, the problems are formulated with nonlinear constraints. This makes it difficult to solve the model. Indeed, there are several local solutions because of the non-convexity of the model. Therefore, the optimal global solution is not guaranteed. As shown in Table 3, authors have used various problem-hniques classified mainly into mathematical and heuristic methods to obtain an optimal solution.

**Table 3.** Solving techniques.

| Solving Method | Optimization's Goal | Contributions | References |
|---|---|---|---|
| The fuzzy logic algorithm has been used for improved prediction accuracy | Used especially in wind energy processes | The proposed stochastic model used for renewables ressource (wind energy) and their market price | [46,53,54] |
| Monte Carlo simulation method | PV radiation prediction (if the problem cannot be solved by mathematic or physical method, it is digitized with repeated random sampling) | Simulation results demonstrate that prediction accuracy increased | [47,48,55] |
| Empirical mode and artificial neural network (ANN) decomposition | Combination of traditional wind turbine fault diagnosis algorithms | Forecasting results have been refined over conventional methods | [55] |
| ANN algorithm | This algorithm is mainly used for an interconnected system | It improves computing efficiency and predictions are more precise | [33,49,56] |
| A geographic routine algorithm based on ant colony optimization | Study of density/efficiency and performance trade-offs | The denser the network, the better the performance up to the saturation limit | [57–59] |
| Genetic and adaptive heuristic search algorithm (based on the evolutionary idea of natural selection) | Multiple DERs reliability problem solving | Optimal sizing using a genetic algorithm | [60–62] |
| Firefly algorithms (FFA) are inspired by firefly that creates a mathematical equation of these behaviors) | Optimization of energy flow, transmission and distribution lines. Create a flexible change of lines according to their efficiencies | An electrical system based on the FFA of choosing the most efficient line | [50,51,63,64] |
| Artificial bee colony | DER placement and sizing process | Placement and sizing of DERs in an electrical system | [65–67] |
| Quantum PSO (MSC quantum particle swarm optimization (QPSO), based on quantum behavior) | To change updating strategy and obtain high searching accuracy | Authors proposed an improved model of traditional PSO | [34,68,69] |
| Particle swarm algorithm | Considering the uncertainty in the optimal energy management | A probabilistic framework for management of microgrid | [70–72] |

### 3.2.1. Mathematical Methods

Many authors employed mathematical methods to ensure optimal management of the energy resources that make up the VPP.

- Fuzzy algorithm: Specially used in wind energy processes. In [53], the fuzzy logic algorithm has been demonstrated to increase prediction accuracy. The authors used a stochastic model to consider the uncertainty of renewable generations and market prices. An iterative procedure has been used in [72] based on the zone-based observation and focusing algorithm, which is divided into two parts. The first part assigns optimal solution area determination, while the second part associates with a local search to obtain the optimal solution. A local search is performed in a second step to obtain the optimal solution. The possibility of obtaining a local optimum with this approach is minimal. Other studies such as [46,54] have also used the same algorithm.

- The authors of the articles [37,73] use a branch-and-bound system that guarantees an intelligent hunt for the optimal outcome. It consists of evaluating the different options based on the value of integer variables, then excluding the combinations that do not respect certain constraints, and finally determining the optimal conditions according to their limits. It facilitates the convergence to the global optimum of the problem, since it has different strategies to explore the field of results. Therefore, its advantage is to limit significantly the search for the optimum. Nevertheless, the major disadvantage of this system is that it is memory intensive, since each possible result

must be independent, so it must contain all the information for the branching process. This also makes it impossible to solve a global structure to obtain the result.

- The authors in [74] decomposed the PPV auction problem into different power demands by using dynamic scheduling. This approach demonstrated good practicability for rebalancing responses to intraday demands with short continuation times. The system has advantages: the first is its ability to handle separate variables, constraints, and queries at the level of each subproblem rather than considering all aspects simultaneously in a full decision model. The second is its ability to increase the efficiency of the resolution by avoiding repeating the exact calculation several times.

### 3.2.2. Heuristic Methods

Heuristic methods can bring reasonable solutions to problems, but they may not offer the best solution. However, it is faster than the pyrolysis method. In addition, another key feature of these methods is that they allow for difficult modeling conditions, which provides greater flexibility. When the problem has many variables, heuristic methods are useful.

### 3.2.3. Summary Methods

As previously mentioned, the model most often described by authors is a mixed integer linear model, which is mainly solved by mathematical methods. The application of branch-and-bound techniques are particularly popular due to their fast convergence to a unique optimal solution. Nevertheless, the successful application of heuristic methods (approximate algorithms) relies on the simple study of models of high mathematical complexity and on obtaining sufficiently robust solutions with a reasonable computation time.

### *3.3. Multiobjective Optimization Algorithm*

Refs. [74–76] and other studies have proposed minimizing (or maximizing) different impact standards at the same time when developing VPP to achieve the best balance between them. The authors in [74] seeks to optimize the benefits of VPPs and minimize the cost of self-consumption of PPVs. Other studies, such as [75], seek to maximize the benefits of VPPs while limiting carbon emissions. Meanwhile, the authors in [76] propose to simultaneously maximize the economic benefits of VPPs and decrease the economic risk of VPP participation in the electricity market.

### *3.4. Distributed Optimization Algorithm*

There are essentially three types of decentralized methods:

- Iterative method based on information exchange [77]: This method involves a centralized information coordinator and a small number of regional controllers. These controllers make their decisions individually after receiving the incentive or control signals from the information coordinator. After iterations between the two, the final decisions of the regional controllers converge based on specific criteria.
- Game theory method [78]: The Nash equilibrium is achieved in a fully distributed manner in this method. Participants adopt tactical or selfish strategies, and they are free to cooperate or not to cooperate.
- Auction-based method [79]: In this method, participants can exchange energy in both directions according to the established rules. To solve the trust problem between the participants, blockchain technology is implemented, and a sensible smart contract is required.

Unlike reality, complete information is required for the existing decentralized control strategies, and it is assumed that the participants are completely rational during the decision-making process [80]. Furthermore, the coupling relationship between the coordinator and the controlled objects is always strong, even in the decentralized structure, which will lead to many difficulties in the design and implementation of VPP.

In [80], the authors created a blockchain-based VPP energy management platform by developing a distributed energy exchange algorithm and blockchain implementation.

They modeled energy exchange and grid services for home customers with a variety of loads, energy storage, and local renewables in particular. Users might potentially engage with one another to swap energy and utilize the VPP to deliver aggregated network services. They created a distributed optimization method to manage the customers' energy schedules and network services because they are all autonomous. They have created a blockchain prototype for VPP energy management. Finally, they used the blockchain technology to implement their algorithm.

The authors of [35] suggested a decentralized, privacy-preserving flexibility planning framework that allows households' net load plans to be coordinated to improve a community microgrid's operational efficiency. The paradigm enables bottom–up flexibility coordination by weighing prosumers' individual techno-socio-economic aims for flexible resource usage (including reserve service for frequency containment) against the community's common goal of lowering peak demand. The authors discovered that design elements had a considerable impact on flexible scheduling's socio-technical performance metrics.

In [81], the authors presented a new bottom–up approach to face the challenge of real-time flexibility utilization by taking into account a large number of complex and heterogeneous DERs with time-varying states.

## 4. Electricity Markets

The liberalization and opening of electricity markets to competition have already been implemented in most countries. The main objectives of this liberalization strategy are to increase the economic effectiveness of the operations of electric companies, to finance new investments in electric infrastructure, and to lower the ultimate prices of electricity supply. This change in the electricity sector has resulted in the independent operation of generation, transmission, distribution, and retail activities instead of a vertical structure where all activities were integrated. Liberalization began toward the close of the twentieth century, and the bulk of power markets were formed around a temporary wholesale market. Other markets, such as Texas, have campaigned for bilateral transactions instead of a centralized pool for all energy trade. In developed power markets, day-ahead, forward, and futures markets are indeed available, providing for price portfolio diversification when buying and selling electricity. In addition, today's electricity system is characterized by a very large development of renewables that increase the need for additional balancing mechanisms due to deviations in the generation schedule of renewable sources. The participation of VPPs in various electrical markets is discussed in this section.

### 4.1. Futures and Forward Market

This market allows the procurement of a quantity of energy through contracts to buy and sell firm energy at a fixed price for a set term. These markets are usually tradable on a standardized exchange, while the forward markets are self-regulated. The main advantage of participating in this market is that the VPP avoids the risks derived from price volatility in the day-ahead electricity market. The capability of VPP power trading in the forward market has been investigated in [36,73,82]. In addition, the VPPs can exploit the arbitrage opportunity between the day-ahead and markets to maximize their profits in [36,82].

### 4.2. Bilateral Contracts

Bilateral contracts are direct agreements between a producer and a consumer based on a number of factors (price, volume of energy delivered, contract period, and minimum power to be supplied/consumed, among others). The benefit of this sort of contract is that it eliminates pricing uncertainty and thus ensures long-term price stability, making both power generation and consumer industry processes more lucrative. In the reviewed literature, articles [19,37,83–88] presented a VPP model in which a bilateral contract must deliver half or all of the demand in a one-week time frame. Due to the volatility of the

market price and the probable restrictions of the transmission system operator, this contract gives a good possibility to guarantee the VPP's revenue.

### 4.3. Day-Ahead Market

The day-ahead market allows electricity transactions for each hour of the next day through an offering mechanism by market participants. VPPs allow small-scale prosumers access to electricity markets for the sale of their power generation. As a result, production facilities tend to generate as much energy as possible in order to optimize the VPP's operational profit. Therefore, the research articles take into account the VPP's participation in the day-ahead electricity market to maximize operational profit [15,83,89,90]. In the electrical markets, the VPP is primarily a buyer, but in some of the work evaluated, the VPP is a decision-maker [15,17,38,79,91–97]. This feature is beneficial, since auction results can affect day-ahead power market pricing in the VPP's favor.

### 4.4. Reserve Market

To ensure demand coverage and security of supply, the reserve market uses a system of increased generation reserves. Generator bids are generally remunerated at a marginal rate. The expansion of non-dispatchable renewables (primarily wind and photovoltaics) has significantly reduced power reserve margins in the power system, making this technique increasingly critical. Various works in the literature [13,29,32,39,84,98–107] offered various approaches for the VPP to make optimum judgments in the day-ahead and reserve power markets in order to maximize economic profit and provide acceptable levels of network security. The results obtained showed that the reserve market is more significant in periods of maximum demand, as an unforeseen event can impact the situation more. In addition, when creating additional renewable energy, the VPP has an incentive to selling energy into the day-ahead market or refilling storage systems rather than engaging in the reserve market. The profit linked with this market for VPP does not rise in a consistent manner. Energy and reserve markets are cleared independent of certain electrical markets, such as the Iberian power market in Spain. It should be noted that the markets indicated are cleared at the same time in other places, such as the California Independent System Operator in the United States.

### 4.5. Intraday Market

The main objective of the intraday market is to match the energy exchanged in the day-ahead market with higher accuracy, because there is more information in this session. The increase in renewable energies and their unpredictability make it essential to correct offers and adjust imbalances before real time. As a result, intraday markets are becoming increasingly important. Moreover, this market can also benefit the agents who participate in it. If a set of generators goes down, for example, the agents can consider purchasing the energy they sold in the prior day-ahead market during an intraday session. Studies [40,79,90,91,108–114] include the marketing of VPP energy in intraday markets to increase profits. Participation of VPP in the intraday electricity market has been investigated in [40,79,90,91,108–114].

### 4.6. Real-Time Market

An imbalance in trading power can still occur as the dispatch hour approaches despite the intraday markets allowing VPPs to make planned energy adjustments after the day-ahead market. As a result, the real-time market is the last chance to strike a balance between production and demand. The articles' objective is to minimize the imbalance error and associated cost. VPPs' access to these markets is crucial for keeping the balance of generation and consumption due to the fluctuation of renewable energy sources.

## 5. Practical Implementation of a Virtual Power Plant

Various VPP initiatives have been established across the world to improve energy management by taking into account various types of RESs and levels of consideration (i.e., household, HAN, and grid levels) in order to accomplish technical, economic, or combined techno-economic goals. The first genuine use of virtual power plants (VPPs) was created in Europe between 2001 and 2005, with an emphasis on fuel cells. VFCPP's major purpose was to install fuel cells as a renewable energy source in homes, allowing utilities to regulate CHP generators during peak hours to optimize profit for both the user and the utility. Because VFCPP was created using MILP optimization and only considered fuel cells, adding additional types of resources makes the process extremely complicated and sluggish, which is not efficient. Smartpool project from the company Next Kraftwerke (2015) is one of important projects implemented in Germany. The implemented VPP manages more than 2900 DERs through the concept of the cloud. The EDISON project uses electric vehicles as mobile electricity storage units to solve Denmark's wind power imbalance. The data-gathering system, controller, and communication module all play important roles in this project's scheduling. This study used a heuristic optimization technique to detect the flow of energy direction and quantity at 15-min intervals. However, due to the nature of other forms of Res and the requirement for short interval time, the proposed algorithm is unable to meet combined techno-economic goals at a higher level of consideration.

The FENIX project integrates solar, wind, and CHP units in two separate situations, covering northern and southern scenarios, using both technical and commercial VPPs. CHP and PV are combined in a low-voltage small-scale network in the northern scenario. Although the heuristic optimization approach used in this project enables real-time energy management, it is incapable of dealing with a large-scale network with several voltage levels. In addition, the southern scenario concentrated on generators in a medium-scale grid with the goal of optimizing day-ahead demand. Another initiative created in the Netherlands from 2005 and 2007 is the Power Matcher, which provides a market mechanism. The maximum number of CHP units in this project is 25 in a microgrid with 25 homes. The Web2Energy project, which was carried out in Germany and Poland, involved aggregating a limited number of energy units in 200 houses with the goal of providing smart metering, RES aggregation, and remote control and automation. The POSITYF project (2021), which is being implemented in Spain, France, Switzerland, and Germany, dynamically coordinates the energy and capacity of DERs to provide auxiliary services to the system. The notion of a dynamic virtual power plant is also developed in this research (DVPP). The POSITYF project, on the other hand, leverages the cloud to regulate DERs across several PCCs with the grid. The VPP solely controls renewable sources (dispatchable and non-dispatchable) and does not consider storage in some simulation situations.

The value of the VPP global market is predicted to exceed one billion dollars in 2023 [13]. The high penetration of RESs and the rising number of DG units make this upward trend faster than before.

The practical projects analyzed have demonstrated the viability and great potential of this technology. However, virtual power plants are especially vulnerable to cyber attacks due to three key factors inherent to the technology: Internet of Things (IoT), cloud computing, and the accessibility of the physical location of the hardware. The following are the goals of assuring cybersecurity in VPPs: information availability and reliability, integrity protection and safety against improper alteration, and confidentiality and proprietary data security [115].

## 6. Conclusions and Future Directions

The objective of this review was to analyze the state of the art of VPP. This article begins by defining the difference between a microgrid and a VPP. Then, we conducted a detailed study of the VPP concept and its architecture. This review classifies and analyzes about 100 research studies in this area according to the definition of the main objective, the problem formulation, the solution method, the selection of the solution method, the

involvement with different electricity markets, and the application of the proposed VPP model to real case studies. This review evaluates the contribution of each aspect analyzed to provide helpful knowledge for further research.

We were able to identify these conclusions:

- In order to achieve optimal control and coordination between components and thus maximize operating profit, researchers have focused on developing VPP models. There is a wide variety of these models.
- The models developed are becoming complete and complex and include more operating constraints. Moreover, more advanced optimization techniques are required to reach an optimal solution.
- The decentralized generation in the VPP has contributed to more active participation in different types of markets; we have noticed the inclusion of bilateral contracts, forward contracts, balancing markets, and the day-ahead spot market.
- The proposed models have rarely been applied to real cases, as in industrial processes that require the management of electricity consumption and its production facilities.

This comprehensive review will be a knowledge base for all researchers in this field. Our study and research perspective is to develop a Multi Agent System (MAS) for advanced distributed energy management of a CVPP. The concept concerns CVPP and distribution units' optimal operation and hence utilizes local intelligence and communication technology via the MAS technique. MAS does not rely on the central decisions and can take proper steps according to the environmental changes.

**Author Contributions:** Conceptualization, W.N.-T. and S.B.E.; methodology, S.B.E. and M.B.; formal analysis, W.N.-T.; investigation, W.N.-T.; writing—original draft preparation, W.N.-T.; writing—review and editing, S.B.E., E.H.-F. and M.B. All authors have read and agreed to the published version of the manuscript.

**Funding:** This work has been carried out in the framework of the European Union's Horizon 2020 research and innovation program under grant agreement No 957852 (Virtual Power Plant for Interoperable and Smart isLANDS'-VPP4ISLANDS).

**Conflicts of Interest:** The authors declare no conflict of interest. The founding sponsors had no role in the design of the study; in the collection, analyses, or interpretation of data; in the writing of the manuscript, and in the decision to publish the results.

### Nomenclature

The following abbreviations are used in this manuscript:

| | |
|---|---|
| VPP | Virtual Power Plant |
| TVPP | Technichal Virtual Power Plant |
| CVPP | Commercial Virtual Power Plant |
| DERs | Distributed Energy Resources |
| RES | Renewable Energy Resources |
| DSO | Distribution System Operator |
| TSO | Transmission System Operator |
| ISO | Independent System Operator |
| WT | Large-Scale Wind Turbine |
| PV | Centralized Photovoltaic Station |
| DAM | Day-Ahead Market |
| RBM | Real-Time Balancing Market |
| ASM | Ancillary Service Market |

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
