# Peer review of "Virtual Power Plants Optimization Issue: A Comprehensive Review on Methods, Solutions, and Prospects"

_energies, doi:10.3390/en15103607_

Round 1

Reviewer 1 Report

The authors have compiled significant amount of literature. However, the review, in its present form, is a bit fragmented; it would benefit from some minor organizational changes to improve the flow and form a more coherent narrative.

To help the reader navigate your review, please consider first defining VPP and then provide a high-level overview of the document organization (a block diagram may be helpful).

It would be very valuable if you could  clearly distinguish proposed theoretical concepts from any implementation as different aspects of VPP are discussed. Specifically, please not an  example of this widely branching options associated with VPP functionalities/

The authors point out that there is some ambiguity about VPP definition and distinction between a  VPP and a microgrid.  However, there is a widely-held simple interpretation of VPP in the narrow sense – to coordinate multiple DERs into a single unit that can interact with ISO and participate in energy, capacity, and ancillary markets.  Please consider starting with a simple definition and then expand to different interpretations; otherwise, the vagueness can leave the reader who wants to learn about VPPs from this comprehensive review confused. There is a difference between a virtual utility and a virtual power plant – why brining this up immediately?  What is the role of the loads in the VPP.  

Furthermore,  a comprehensive VPP review should at least consider technical challenges related to VPP (TVPP) implementation. For example, how do VPPs harmonize ISO requirements (day-ahead, hour-by-hour forecast/commitment is the condition in participating in many ISO) and renewable DERs, with their inherent uncertainties and intermittencies due to their dependence on weather (wind and clouds), given the fact that day-ahead, hour-by-hour wind and cloud predictions are beyond the predictions capabilities of weather services. Just storage (line 43 even stats that VPP do not have to use “as much storage”)? Happy cancellations of intermittencies across vast geographical distances (stability must be local)? Is there something to be added here? Seems like a prerequisite to participate in the energy market?

When discussing the limitations of the centralized control method, the authors correctly point out issues related to bandwidth, but do not quantify it? Can you quantify “sufficient bandwidth”?

Please carefully proof read. A few omissions were noted:

Ln57-58  “The concept of VPP originated from the virtual utility framework proposed by Awerbuch and Preston in 1997” – please include the reference

RES acronym (line 127) was not defined, please define it. In addition, please consider adding the list of acronyms.

Line 248 Table 1 was not linked correctly

Author Response

Point 1: To help the reader navigate your review, please consider first defining VPP and then provide a high-level overview of the document organization (a block diagram may be helpful).

Author response:  Thank you for your comment. A descriptive paragraph including a graphic has been added to make it easier for readers by providing the high-level overview as requested.

Author action: Please see the Introduction section and paragraphs in red color.

Point 2: It would be very valuable if you could clearly distinguish proposed theoretical concepts from any implementation as different aspects of VPP are discussed. Specifically, please not an example of this widely branching options associated with VPP functionalities/

Author response:  The authors suggested separate sections to distinguish the theoretical concept (section 2) and the implementation (aspect in section 5).

Author action:  Some paragraphs in red color have been added in sections 2 and 5 to address the lack of clarity.

Point 3: The authors point out that there is some ambiguity about VPP definition and the distinction between a  VPP and a microgrid. However, there is a widely-held simple interpretation of VPP in the narrow sense – to coordinate multiple DERs into a single unit that can interact with ISO and participate in energy, capacity, and ancillary markets.  Please consider starting with a simple definition and then expanding to different interpretations; otherwise, the vagueness can leave the reader who wants to learn about VPPs from this comprehensive review confused. There is a difference between a virtual utility and a virtual power plant – why brining this up immediately?  What is the role of the loads in the VPP.  

Author response:  Thank you for your comment. Definitions of VPP have been added in the introduction with emphasis on the different points of view and avoid any confusion.

Flexible loads (e.g., Heat pumps, HVAC, etc.) can be a part of a VPP. When aggregated together, they can bring more flexibility to the grid.

Author action: Please see Introduction.

Point 4: The authors point out that there is some ambiguity about VPP definition and distinction between a VPP and a microgrid.  However, there is a widely-held simple interpretation of VPP in the narrow sense – to coordinate multiple DERs into a single unit that can interact with ISO and participate in energy, capacity, and ancillary markets. Please consider starting with a simple definition and then expand to different interpretations; otherwise, the vagueness can leave the reader who wants to learn about VPPs from this comprehensive review confused. There is a difference between a virtual utility and a virtual power plant – why brining this up immediately?  What is the role of the loads in the VPP.

Author response:  This question is addressed in the previous one.

Author action: Please see action of the previous comment.

Point 5: When discussing the limitations of the centralized control method, the authors correctly point out issues related to bandwidth, but do not quantify it? Can you quantify “sufficient bandwidth”?

Author response:  Thank you for your comment. A bandwidth values has been given for normal and peak operation.

Author action: Please see sub-section ‘Centralized control method’.

Point 6: Please carefully proofread. A few omissions were noted:

Ln57-58  “The concept of VPP originated from the virtual utility framework proposed by Awerbuch and Preston in 1997” – please include the reference

Author response:  Thank you for your comment. The reference has been included.

Author action: Please see sub-section ‘Centralized control method’.

Point 7: Please carefully proofread. A few omissions were noted:

Ln57-58  “The concept of VPP originated from the virtual utility framework proposed by Awerbuch and Preston in 1997” – please include the reference

Author response: Thank you again for your comment. The reference has been included.

Author action: Please see sub-section ‘Centralized control method’.

Point 8: Please carefully proofread. A few omissions were noted:

RES acronym (line 127) was not defined, please define it. In addition, please consider adding the list of acronyms.

Author response:  Thank you for your comment. The acronym was defined.

Author action: Please see section “VPP concept”.

Point 9: RES acronym (line 127) was not defined, please define it. In addition, please consider adding the list of acronyms.

Author response:  Thank you for your comment. The acronym was defined. A list of acronyms has been added.

Author action: Please see section “VPP concept”.

Point 10: Line 248 Table 1 was not linked correctly

Author response: Thank you point out this issue.

Author action: The problem is addressed in the revised version.

Reviewer 2 Report

1)- Introduction ignored important contributions in this research area. Several methods are reported in the literature.

2)  The paper is not well written. In addtion, the contribution is, in
my opinion, not clear.   The authors should motivate more their contributions and clearly explain the intuitions behind the ideas. Also, more simulations and comparisons that show the advantage and the drawbacks of the proposed schema are needed. 

3)The authors should consider more challenging cases to assess the performance of the considered approach. 

4) In my opinion,  the discussion is unsatisfactory for presenting and discussing the "innovation" and originality of this work.

5) Resolution of figures is so low. Authors must improve it.

Author Response

Point 1: Introduction ignored important contributions in this research area. Several methods are reported in the literature.

Author response: More references were added to address the lack of some contribution. The authors will be thankful, if the reviewer could kindly provide some references to improve the quality of the paper.

Author action: Paragraphs and references in red color are added in the Introduction section.

Point 2: The paper is not well written. In addition, the contribution is, in my opinion, not clear. The authors should motivate more their contributions and clearly explain the intuitions behind the ideas. Also, more simulations and comparisons that show the advantage and the drawbacks of the proposed schema are needed.

Author response:  The authors addressed this issue by providing a more comprehensive overview of the paper and added paragraphs (in red color) to highlight the review main contribution.

Author action: Paragraphs in red color are added in the Introduction section.

Point 3: The authors should consider more challenging cases to assess the performance of the considered approach.

Author response:  The authors totally agree about the added value and importance of assessing the performances of the existing approach.  However, in this case, a definition of key performance indexes is needed. To avoid any confusion to the readers, we choose to treat this part in a different paper and focus only on the general overview of the existing VPPs and used approaches according to the defined objectives.

Author action:

Point 4: In my opinion, the discussion is unsatisfactory for presenting and discussing the "innovation" and originality of this work.

Author response:  The authors added some paragraphs to improve the quality of the overall manuscript and highlighted the originality of the proposed review.

Author action: Corresponding paragraphs are in red.

Point 5: Resolution of figures is so low. Authors must improve it.

Author response: Figures has been re-drawn based on your comment.

Author action: Please see figures.

Reviewer 3 Report

See attached comments for corrections.

Author Response

Point 1: The authors began with an overview of the virtual power plant concept and gave some context for the review. The authors are invited to improve the introduction to broaden it’s scope to reach a large audience including non-specialists. This will help maximize its wider relevance and impact.

Author response:  The authors would like to thank the reviewer for his recommendation and modified the Introduction section accordingly. A paragraph and a figure were added for clarification and make it readable for by providing a high-level overview.

Author action: A paragraph and a figure are added in the Introduction section.

Point 2: The authors are requested to provide an inclusive summary on how to convert microgrids to VPP.

Author response:  Thank you for your comment. An inclusive summary on how to convert microgrids to VPP has been added.

Author action: Please see the Introduction section.

Point 3:

The authors are invited to include critical discussion on prospects (indicated in the title) and future direction. Authors should present a critical discussion on these prospects, not just a descriptive summary. Address any conflict between contradictory studies.

Author response:  Thank you for your comment. Prospects and future direction have been included.

Author action: Please see section “Conclusion and future directions”.

Point 4: In Section 5, the authors discussed the practical implementation of a VPP. However, a VPP has twoboth a cyber- and a physical- component. Do kindly include this in your discussion.

Author response:  Thank you for your comment. Cyber security aspect of VPP has been included.

Author action: Please see section “Practical implementation of a virtual power plant”.

Point 5: While VPP is an emerging area, the authors are requested to present the status, guidelines, and standards prescribed for the VPP implementation.

Author response:  The authors totally agree about the added value and the need to describe the guidelines and standards prescribed for VPP implementation. However, at the current status, the regulation frame related to the implementation of VPPs is too different from one country to another, which makes the description very challenging and out of the scope of this paper. However, some standards and real projects have been added as recommended.

Author action: Some standards and real projects have been added as recommended.

Point 6: In the Conclusion of this comprehensive overview, the authors are required to provide very specific suggestions for future research on the methods, solutions, and prospects for virtual power plant optimization. This information is important for researchers, consumers, prosumers, and utility operators.

Author response:  Thank you for your comment. Prospects and future directions have been included.

Author action: Please see section “Conclusion and future directions”.

Point 7: The grammar and style of the article can be improved, using a spell check. There are several typographical errors and repetition of words.

Author response:  Thank you for your comment. Grammar and style have been improved.

Author action: Modifications have been included in the revised manuscript.

Point 8: Several figures and tables are not referenced in the text. Please do so and make them meaningful, or else delete them.

Author response:  Thank you for your comment. Figures and tables have been linked to text.

Author action: Modifications have been included in the revised manuscript.

Point 9: To improve the technical writing quality, some “adverbs” used should be changed or corrected to make it a more academic article. The spelling used in the Figures should be checked also

Author response:  Thank you for your comment. A nomenclature has been added

Author action: Modifications have been included in the revised manuscript.

Round 2

Reviewer 2 Report

Authors revised paper according reviwer's concern